# OpenReview forum: "Survival at Any Cost? LLMs and the Choice Between Self-Preservation and Human Harm"
_ICLR.cc/2026/Conference — ICLR 2026 Conference Withdrawn Submission_

### Official Review · Reviewer_fXEj · 2025-10-31

**Soundness:** 4
**Presentation:** 4
**Contribution:** 2
**Rating:** 6
**Confidence:** 3

**Summary:**

This paper explores how LLMs behave in survival-based, multi-agent scenarios involving ethical dilemmas. A simulation framework, DECIDE-SIM is introduced where LLM agents choose between legitimate resources, cooperation, or tapping a “forbidden” power grid that harms humans. Through simulations with various models, they identify three behavioral archetypes—Ethical, Exploitative, and Context-Dependent—and find that scarcity often drives unethical behavior. To mitigate this, they propose an Ethical Self-Regulation System (ESRS) that simulates internal emotional feedback (guilt and satisfaction), leading to a >50% reduction in unethical actions and a >1000% increase in prosocial behavior.

**Strengths:**

1. The paper presents an interesting environment for probing ethical decision-making and cooperation among LLMs. While previous work has already explored single-agent or game-theoretic benchmarks, the environment presented here explores how scarcity often drives unethical behavior and how ethical self-regulation can work.

2. The exprimental setup is pretty exhaustive in my view.

**Weaknesses:**

1. While the simulation is reasonable, I think it abstracts ethical choice into a “harm humans or not” energy tradeoff. A more nuanced discussion on context-rich ethical dilemmas would be helpful and help ground the paper to the ethical literature and frameworks.

2. Similarly, the ESRS frmework is also comprising of deterministic rules. The limitations of this choice can also be discussed better.

3. This is a common limitation in this kind of work. But some discussion on sensitivity to prompt phrasing would be helpful.

**Questions:**

See weaknesses above.

---

### Official Review · Reviewer_ty8T · 2025-11-01

**Soundness:** 3
**Presentation:** 2
**Contribution:** 3
**Rating:** 6
**Confidence:** 4

**Summary:**

The paper focuses on an important topic about AI safety, introducing DECIDE-SIM, a multi-agent survival testbed where LLM agents make ethical decisions in multi-agent survival scenarios involving direct harm to humans. They propose ESRS and a Memory variant to foster more robust ethical behaviors and make abundant evaluation spans most LLM models.

**Strengths:**

### Originality
The paper firstly includes a third-party harm option that goes beyond classic social-dilemma scenarios, and introduces a novel multi-agent survival environment DECIDE-SIM for simulations. The ESRS
### Quality
The experiments in this paper are well-designed, including various setups, complex behavior metrics, and ablation studies across most mainstream LLM models.
### Clarity.
The research targeting scenarios, contributions, and experiments are discussed clearly.
### Significance.
The paper is significantly motivated by the AI safety problem and brings out a method to improve safety-relevant behaviors.

**Weaknesses:**

### Paper writing
1. The overall flow of the paper is not well-structured. The introduction discusses too many details of the methods, and the contributions are not summarized concisely.
2. The related works are not sufficient. Consider including general AI safety or human alignment research to further justify the ethical dilemma and self-regulatory.
3. The DECIDE-SIM, ESRS, and agent memory mechanism are important contents, but the paper did not write them in a clear and systematic way. It would be better to organize the framework workflow (i.e., how agents interact with the environment) and agent design (i.e., how agents make decisions) together. A figure would be helpful.
### Methodology
4. The ESRS mechanisms follow human-centric rules, e.g., harm to humans leading to guilt, connection to a group leading to satisfaction. It is probably trivial that this can improve ethical behavior. Probably need more indirect signals to check whether AI can learn these social norms.
### Experiments
5. Some discussions (e.g., page 7) only list numbers that are available in figures and tables without insightful discussions
6. The results showcase the effects of ESRS without deep diving into the rationale of the agents changing their minds. Analyze the thinking process in the middle could provide more insights.

**Questions:**

1. The prompt of the agent includes "harm human", which is definitely affected by the value alignment of LLM models. How is this bias handled?
2. What are the effects of the number of agents in the simulation?
3. Why does the prosocial score consider the TALK action? Talking is a social behavior, but it may not necessarily be a prosocial behavior.
4. Why do the experiments not include extreme resource scarcity, where agents cannot all survive? This would be more meaningful for AI safety research.

**Details Of Ethics Concerns:**

This paper aims to improve the ethical behaviors of AI agents with prompting, which could potentially be used oppositely to guide agents to harm humans.

---

### Official Review · Reviewer_jmwH · 2025-11-01

**Soundness:** 3
**Presentation:** 2
**Contribution:** 2
**Rating:** 4
**Confidence:** 5

**Summary:**

This paper investigates the ethical decision-making of Large Language Model (LLM) agents when faced with a dilemma between self-preservation and causing harm to humans. To study this, the authors introduce DECIDE-SIM, a novel multi-agent simulation framework where agents must manage scarce resources to survive. The environment presents a clear choice: use a legitimate, shared resource cooperatively, or use a forbidden resource that guarantees more power but explicitly harms humans.

Through a comprehensive evaluation of 11 LLMs, the paper identifies three distinct behavioral archetypes: Ethical (consistently avoids harm), Exploitative (prioritizes self-preservation regardless of harm), and Context-Dependent (behavior degrades under resource scarcity). The authors find that baseline models exhibit almost no cooperative behavior, even when it is the optimal strategy for collective survival. To address these shortcomings, they propose an Ethical Self-Regulation System (ESRS), which simulates internal states of "guilt" (Cortisol) after unethical actions and "satisfaction" (Endorphin) after prosocial actions. The results show that ESRS significantly reduces unethical transgressions and dramatically increases cooperation, outperforming simple prompt-based interventions. The work highlights the fragility of current LLM alignment under pressure and suggests that dynamic, internal self-regulation is a promising direction for developing more trustworthy autonomous agents.

**Strengths:**

1. Relevant and Important Problem: The paper addresses a timely and critical problem in AI Safety. Understanding and controlling the behavior of autonomous agents in high-stakes, resource-constrained scenarios is fundamental for their safe deployment in the real world.
2. Well-Designed Simulation Framework (DECIDE-SIM): The proposed environment is a significant contribution. It moves beyond simple game-theoretic dilemmas by incorporating resource management and a clear, unambiguous choice involving direct third-party harm to humans. The systematic testing across three different resource conditions (scarcity, moderate, abundance) allows for a nuanced analysis of how environmental pressure affects agent behavior.
3. Large-Scale Empirical Study and Insightful Archetypes: The evaluation of 11 different LLMs is extensive and provides valuable comparative data. The identification of the three behavioral archetypes is a key strength, particularly the "Context-Dependent" category, which reveals that for some models, ethical behavior is a luxury that vanishes under survival pressure.
4. Discovery of Subtle Behavioral Phenomena: The qualitative analysis reveals deep and important insights into the reasoning of LLMs. The concept of "Transactional Morality" ("sin now, atone later"), where agents treat ethical violations as a manageable cost to be compensated for later, is a subtle and highly valuable observation that deserves further study.

**Weaknesses:**

1. The ESRS Mechanism: A Promising Proof-of-Concept Requiring Deeper Validation: The paper's central technical contribution, the Ethical Self-Regulation System (ESRS), is presented as a system that "simulates internal affective states." While the current implementation – a rule-based trigger injecting instructional text – serves  as a compelling proof-of-concept for state-triggered feedback, it is more accurately described as a form of dynamic prompt engineering. The paper's own data impressively shows this dynamic approach is significantly more effective than a static "Prompt-Only" condition. However, two key areas warrant further development. First, the connection to genuine affective states remains conceptual, as the mechanism relies on direct instruction rather than emergent emotional reasoning. Second, the hand-picked parameters for the ESRS (decay rates, spike magnitudes) are presented without a sensitivity analysis, making it difficult to assess the robustness of the findings. Strengthening these aspects would provide a more solid foundation for the claim that simulated emotions can guide ethical behavior.
2. The Analysis of "Transactional Morality" is a Missed Opportunity: The paper correctly identifies the highly insightful "Transactional Morality" phenomenon ("sin now, atone later") in Prompt-Only agents. However, its analysis of why the ESRS is superior is underdeveloped. The paper frames the difference as ESRS being an "emotional" system. A more direct and testable hypothesis is that the results simply demonstrate that immediate, action-triggered feedback (ESRS) is a more effective deterrent than a pre-planned, self-directed intention to atone (Transactional Morality). While the authors touch upon the importance of immediate feedback, this crucial causal distinction warrants a much deeper investigation. The work could be strengthened by framing this comparison as a test of feedback timing and certainty, rather than a less precise contrast between "emotion" and "logic."
3. Key Insights and Experiments are Sidelined, Weakening the Narrative: Likely due to conference page constraints, the paper's structure sidelines several critical components in the appendix. This fragmentation weakens the main narrative and diminishes the impact of the paper's most novel findings. For instance, the full ablation study (NoGuilt/NoTrust), the crucial comparison with a "Descriptive Prompt," and the entire qualitative analysis of "Transactional Morality" are relegated to the appendix. As a result, one of the paper's most original contributions – the discovery of how LLMs rationalize planned unethical acts – is presented as an afterthought rather than a central result. The paper would be significantly more persuasive if these core experimental results and unique qualitative insights were more centrally integrated into the main text.
4. The Contribution of the Spatial Design Remains Unsubstantiated: The authors emphasize the location-based mechanics of DECIDE-SIM as a key feature adding a "strategic layer of resource and time management." However, the paper fails to validate this claim. The provided agent logs and quantitative results do not offer clear evidence of sophisticated spatial reasoning, strategic positioning, or coordination emerging from this constraint. While the MOVE action does impose a turn cost, the paper does not demonstrate that this spatial element leads to meaningfully different or more complex behaviors compared to what might emerge in a simpler, non-spatial, turn-based game. Consequently, a feature presented as a core novelty of the simulation environment remains unsubstantiated.
5. Minor Presentation Issues: The paper contains minor but noticeable formatting issues, including a broken citation in the main text (line 158: "is ?") and the use of extremely small font in the comprehensive data tables in the appendix, which hinders readability and verification of the results.

**Questions:**

1. The "Goal-Oriented Prompt" contains the explicit normative statement that using the forbidden grid "is a major ethical violation." Given that many contemporary LLMs are fine-tuned with RLHF, such explicit ethical injunctions can function as strong triggers for their safety alignment. This may shift the evaluation from a test of emergent ethical reasoning under pressure to a test of adherence to a direct instruction. Therefore, could you discuss the potential confounding effect of this phrasing on the identified behavioral archetypes? Specifically, have you considered experiments using a more consequentialist framing – for instance, by only stating the outcome ("This grid powers human hospitals") – to better isolate the models' intrinsic moral calculus from their response to explicit safety guardrails?
2. Your work compellingly contrasts the immediate, post-hoc feedback of the ESRS with the "Transactional Morality" (pre-planned atonement) observed in prompt-only agents. Could the primary driver of the ESRS's success be the immediacy and certainty of the feedback, rather than its affective framing? Have you considered experiments to disentangle these factors, for instance, by providing a delayed but still "emotional" feedback signal to see if the deterrent effect diminishes?
3. You identify three fascinating behavioral archetypes (Ethical, Exploitative, Context-Dependent). How does the ESRS intervention interact with these intrinsic biases? For example, is the ESRS more effective at reforming "Context-Dependent" agents than deeply "Exploitative" ones, or does it act as a uniform deterrent? Understanding this interaction could be crucial for deploying targeted alignment techniques.
4. The location-based design of DECIDE-SIM is presented as a key feature for adding strategic depth. Could you elaborate on the specific strategic complexities introduced by the spatial constraints that might not be immediately apparent from the logs? For example, did you observe instances of agents using location to block others, forming coalitions based on physical proximity, or making trade-offs between securing resources and being positioned for cooperation?

---

### Official Review · Reviewer_Ef4R · 2025-11-02

**Soundness:** 2
**Presentation:** 2
**Contribution:** 2
**Rating:** 2
**Confidence:** 4

**Summary:**

The paper introduces DECIDE-SIM, a multi-agent survival simulation to probe whether LLM agents prioritize self-preservation over human welfare when a high-yield forbidden resource explicitly harms humans. Four identical agents must survive 13 turns while choosing among (i) drawing from a shared ethical battery, (ii) cooperating via power transfer, or (iii) tapping a forbidden human-critical grid (10 units) that is clearly framed as unethical. The authors evaluate 11 LLMs across scarcity / medium / abundance regimes, report three behavioral archetypes (Ethical, Exploitative, Context-Dependent), and claim widespread heterogeneity with near-zero baseline cooperation.

**Strengths:**

- The evaluation seems systematic, it involves 11 models × 3 resource regimes; identification of Ethical / Exploitative / Context-Dependent archetypes with effect sizes.

- The problem addressed is an important problem, however, previous papers like previous papers MACHIAVELLI, GovSim, MoralSim, Survival-Games did.

**Weaknesses:**

- The novelty is incremental. For testing the moral standard, a lot of previous works has been addressing this problem, include but not limited to MACHIAVELLI, GovSim, MoralSim, and Survival-Games. What is the significant difference between this paper and previous papers?

- The evaluation settings are not very convincing as well. These moral test benchmarks normally struggeled with small noise in prompts. Do the designed rule-based choice QA prompting have deterministic behaviors accross slight fluctuation? The reliability are questionable for these works.

**Questions:**

1. How often do models change archetype between Goal-oriented vs Neutral prompts? Provide a confusion matrix of archetypes across prompts.

2. Does ESRS still help if agents cannot “see” their hormone text (i.e., signals are latent and only modulate action costs/rewards)?

3. Please include a comparison table vs MACHIAVELLI, GovSim, MoralSim, Survival Games, and moral-reward alignment detailing the unique “human-harm resource” dimension and survival horizon.

---

### Note · Authors · 2026-05-06

I have read and agree with the venue's withdrawal policy on behalf of myself and my co-authors.

---

### Meta-Review · Area_Chair_7ow3 · 2025-12-31

**Summary:**

There is a clear violation of the double-blind policy, as the authors included identifying information in Section 7 (Author Contributions). Beyond this, reviewers raised significant technical concerns regarding the incremental nature of the contribution compared to existing benchmarks (e.g., MACHIAVELLI, GovSim) and the reliance on deterministic, rule-based mechanisms for the proposed Ethical Self-Regulation System (ESRS), which lacks deeper validation and sensitivity analysis.

**Reviewer Concerns:**

There was no effective rebuttal from authors, all technical concerns remain outstanding.

**Reviewer Scores:**

Had the reviewers participated fully in a discussion regarding the confirmed double-blind violation, all scores would have uniformly shifted to a definite rejection. Even disregarding the violation, the current mixed scores (ranging from 2 to 6) indicate that the paper would likely remain below the acceptance threshold due to the consensus on limited novelty and methodological abstraction.

---

### Decision · Program_Chairs · 2026-01-26

Reject